# Choroidal Mast Cells and Pathophysiology of Age-Related Macular Degeneration

**DOI:** 10.3390/cells13010050

**Published:** 2023-12-26

**Authors:** Sara Malih, Yong-Seok Song, Christine M. Sorenson, Nader Sheibani

**Affiliations:** 1Department of Ophthalmology and Visual Sciences, University of Wisconsin School of Medicine and Public Health, Madison, WI 53705, USA; malih@wisc.edu (S.M.); song224@wisc.edu (Y.-S.S.); 2Department of Medical Biotechnology, Faculty of Medical Sciences, Tarbiat Modares University, Tehran 15614, Iran; 3McPherson Eye Research Institute, University of Wisconsin School of Medicine and Public Health, Madison, WI 53705, USA; cmsorenson@wisc.edu; 4Department of Pediatrics, University of Wisconsin School of Medicine and Public Health, Madison, WI 53705, USA; 5Department of Cell and Regenerative Biology, University of Wisconsin School of Medicine and Public Health, Madison, WI 53705, USA

**Keywords:** mast cells, retinal degeneration, inflammation, oxidative stress, fibrosis, choroidal neovascularization, retinal pigment epithelium

## Abstract

Age-related macular degeneration (AMD) remains a leading cause of vision loss in elderly patients. Its etiology and progression are, however, deeply intertwined with various cellular and molecular interactions within the retina and choroid. Among the key cellular players least studied are choroidal mast cells, with important roles in immune and allergic responses. Here, we will review what is known regarding the pathophysiology of AMD and expand on the recently proposed intricate roles of choroidal mast cells and their activation in outer retinal degeneration and AMD pathogenesis. We will focus on choroidal mast cell activation, the release of their bioactive mediators, and potential impact on ocular oxidative stress, inflammation, and overall retinal and choroidal health. We propose an important role for thrombospondin-1 (TSP1), a major ocular angioinflammatory factor, in regulation of choroidal mast cell homeostasis and activation in AMD pathogenesis. Drawing from limited studies, this review underscores the need for further comprehensive studies aimed at understanding the precise roles changes in TSP1 levels and choroidal mast cell activity play in pathophysiology of AMD. We will also propose potential therapeutic strategies targeting these regulatory pathways, and highlighting the promise they hold for curbing AMD progression through modulation of mast cell activity. In conclusion, the evolving understanding of the role of choroidal mast cells in AMD pathogenesis will not only offer deeper insights into the underlying mechanisms but will also offer opportunities for development of novel preventive strategies.

## 1. Introduction

Age-related macular degeneration (AMD), particularly its dry form, is a leading cause of irreversible blindness in older individuals with a prevalence of 0.44% globally [1]. AMD is classified into two forms: dry and wet AMD. Dry AMD, the focus of this review, is characterized by geographic atrophy (GA) of the central retina in the late stages [1]. Dry AMD has a multifactorial etiology and is driven by genetic, environmental, and lifestyle factors [2,3]. The clinical manifestation of AMD encompasses the buildup of drusen, the loss of retinal pigment epithelium (RPE), and photoreceptor cells leading to a gradual loss of central vision [2,3]. Despite significant advances in our understanding of the disease etiology, the precise molecular and cellular mechanisms underlying the pathophysiology of dry AMD remain incompletely understood [3,4,5]. Thus, gaining deeper insights into the cellular and molecular mechanisms involved in pathogenesis of dry AMD holds significant promise in unraveling the disease’s mechanisms, identifying essential risk factors, and paving the way for development of preventive strategies. This knowledge should also facilitate the development of potential biomarkers for early detection and monitoring of the disease’s progression.

A key aspect of dry AMD pathogenesis, namely the loss of retinal pigment epithelium (RPE) cells, has been the focus of many studies. RPE is a single layer of specialized epithelial cells that support the photoreceptor cells’ integrity and function, as well as providing protective and homeostatic mechanisms against oxidative stress, and angioinflammatory and fibrotic processes. RPE cells experience significant functional changes that promote AMD progression. Notably, alterations in their phagocytic and degradative activities lead to the accumulation of lipofuscin (a metabolic byproduct of the visual cycle), the formation of drusen (extracellular deposits between the RPE and Bruch’s membrane) [6,7], and enhanced oxidative stress and inflammatory processes. The role of oxidative stress and inflammation in the pathogenesis of dry AMD has been widely recognized to lead to cellular damage and death. Chronic inflammation is a key contributor to the progression of AMD by promoting the release of pro-inflammatory cytokines and chemokines, as occurring during aging, further exacerbating oxidative stress and recruitment and engagement of immune cells and inflammatory processes [8,9].

The complement system, a component of the innate immune system, which enhances the ability to clear microbes and damaged cells, has been implicated in dry AMD pathogenesis. Genetic studies have linked variants of various complement genes, such as CFH, C3, and CFI to increased AMD risk [8]. Among the notable players is complement factor H (CFH), a crucial regulator of the alternative complement pathway. CFH operates by inhibiting the C3 convertase enzyme and promotes the decay of C3b, thereby controlling complement activation. A common polymorphism in the CFH gene (Y402H) has been strongly associated with an increased risk of developing AMD [8]. This variant presumably alters CFH function, leading to complement pathway over-activation and subsequent inflammation and tissue damage, a hallmark of AMD progression.

Recent studies have also elucidated the role of the cyclic GMP-AMP synthase (cGAS), a cytosolic DNA sensor, in the pathogenesis of dry AMD. The cGAS DNA sensor plays a major role in sensing the Alu-RNA-induced interferon (IFN) responses and degeneration of RPE cells, in part through sensing mitochondrial DNA. However, knockdown of cGAS had little effect on the DNA-induced production of IFN within RPE cells. The cGAS transcript was not detectable in a line of RPE cells derived from induced pluripotent stem cells, and only detected at very low levels in ARPE-19 cells. Thus, cGAS might not be expressed in RPE cells and is not induced in RPE cells exposed to nucleic acids [10]. In addition to the role of cGAS, the activation of IFN-induced-cell-death-related gene expressions and the alteration of RPE cell barrier function in response to RNA have also been implicated in the pathogenesis of dry AMD.

Activation of the IFN pathway by nucleic acids can induce RPE cell loss. While cell death was not observed in RPE cells treated with nucleic acids, gene expression of key effectors of various types of cell death, including caspase 7/8 (apoptosis), gasdermin-D (pyroptosis), and MLKL (necroptosis), was induced upon RNA stimulation. These results support the hypothesis that DNA sensing in RPE cells may sensitize cells to secondary insults by upregulating cell death effectors during degeneration and disease progression, eventually leading to RPE loss. The effects of IFNs on RPE cell barrier function have also been investigated. IFN-β treatment leads to impaired barrier function, which is prevented using an anti-IFN-β antibody. Consistent with these findings, RNA but not DNA was shown to be capable of impairing transepithelial resistance (TER). These results further support the findings that the IFN response in RPE cells is induced through RNA, but not DNA, sensors.

On chromosome 10q26, there are two significant genes, ARMS2 (age-related maculopathy susceptibility 2) and HTRA1 (high-temperature requirement A serine peptidase 1), whose variants are identified as major risk factors for AMD. The precise mechanism through which HTRA1 influences AMD progression remains to be fully elucidated. HTRA1 is thought to be involved in extracellular matrix remodeling and modulation of TGF-β signaling, which could influence angiogenesis and inflammatory processes. The ARMS2 gene, although less well understood, is speculated to be involved in mitochondrial function and may play a role in cell survival and oxidative stress resistance [11,12,13]. A recent study showed expression of a common ARMS2 variant, associated with AMD susceptibility, in RPE cells resulted in increased oxidative stress [13].

Mitochondrial dysfunction is also recognized as a significant factor in RPE cell dysfunction and pathogenesis of dry AMD. This is rationalized by the outer retinal high metabolic activity making RPE and/or photoreceptor cells vulnerable to mitochondrial oxidative damage. These changes could lead to reduced ATP production, increased oxidative stress, and activation of cell death pathways driving RPE cell death, photoreceptor degeneration, and AMD progression [2]. In addition, altered autophagy in RPE cells, a cellular process that degrades and recycles waste and damaged cellular components, could contribute to RPE dysfunction. Autophagy dysregulation contributes to the accumulation of damaged proteins and organelles inside and outside of RPE cells, leading to cellular dysfunction and death. Lipofuscin bisretinoid accumulation in RPE cells disrupts autophagy, resulting in accumulation of cholesterol and activation of acid sphingomyelinase that impedes autophagosome movement [14].

The area least studied is the impact of choriocapillaris dysfunction in AMD pathogenesis. The relationship between choriocapillaris dysfunction and outer retinal degeneration with AMD is often considered within a broader framework including other ocular tissues. For instance, the choriocapillaris complex, consisting of the choriocapillaris, Bruch’s membrane, and RPE layer, is frequently explored in studies examining the early pathological changes in AMD. Degenerative changes in any of these components can have cascading effects on the others, potentially leading to a vicious cycle of degeneration and dysfunction. The choriocapillaris is a dense network of capillaries located beneath the RPE separated by Bruch’s membrane. Its primary function is to provide nutrients and oxygen to the outer retina, including RPE and photoreceptor cells. Any compromise to its functionality can therefore have serious implications for outer retinal health [15,16]. In the context of AMD, several researchers have recently proposed that the degeneration or dysfunction of the choriocapillaris could precede, contribute to, or even initiate the pathophysiological process of AMD. As AMD primarily affects the macular region of the retina, the health of the underlying choriocapillaris is critical for maintaining the structural and functional integrity of the macula [17,18,19].

In addition, recent investigations suggest a potential interplay between CFH, HTRA1, and modulation of thrombospondin-1 (TSP1) activity enhancing angiogenesis and inflammation. We propose TSP1 is a central player in pathogenesis of AMD whose angioinflammatory functions are impacted by CFH and HTRA-1 high-risk variants. Figure 1 illustrates the proposed impact of HTRA1 and CFH polymorphisms on TSP1 modulation of angioinflammatory processes and pathogenesis of AMD. Furthermore, alterations in production of various ECM proteins, and RPE-Bruch’s membrane interactions, are linked to dry AMD development and could be influenced by changes in the TSP1 level and mast cell activity as detailed below.

TSP1 is a multifaceted matricellular protein with important roles in regulation of ocular angioinflammatory processes. The precise mechanics of how CFH and HTRA1 could modulate TSP1 activities and their implications in AMD pathogenesis are still under investigation. We hypothesize that through binding and modulation of TSP1 activity, CFH and HTRA1 alter the inflammatory and angiogenic balance within the choroid, favoring enhanced angioinflammatory action of choroidal mast cells and macrophages. The CFH variant interferes with the interaction of TSP1 with its receptor CD47 on macrophages, preventing their timely clearance and enhancing inflammatory activities [20]. HTRA1, a protease, cleaves the TSP1, releasing its heparin-binding N-terminal domain with demonstrated proangiogenic activity [21]. In addition, these interactions could potentially affect the extracellular matrix (ECM) integrity and the overall choroidal homeostasis, thus highlighting a complex interplay of genetic and molecular factors that could contribute to AMD [21,22] with TSP1 as a central player modulating angioinflammatory processes. 

Age-related ECM changes, including TSP1, affect RPE cell adhesion, survival, and function, while thickening and calcification of Bruch’s membrane hinder nutrient and waste exchange between the RPE and choroid. Recent studies have highlighted the role of calcium and integrin-binding protein 2 (CIB2) in the pathogenesis of dry AMD. Deficiency of calcium and CIB2 in mice leads to AMD pathologies, including sub-RPE deposits and impaired visual function, due to reduced lysosomal capacity and autophagic clearance, and increased mTORC1 signaling. These findings may have significant implications for the treatment of AMD and other mTORC1-hyperactivity-associated disorders [23]. However, how these changes impact the TSP1 level and choroidal mast cell function remains unknown.

## 2. Choroidal Mast Cells and AMD

Although the precise etiology of AMD remains not fully elucidated, it is well recognized that inflammation and immune system dysregulation play crucial roles in its pathogenesis [24]. Recent studies by Lutty and colleagues have begun to explore the uncharted territory of choroidal mast cell involvement in AMD pathophysiology. Their published studies using human donor eyes began to hint to a possible association between mast cell activity and the initiation and/or progression of AMD [25,26,27]. Mast cells are a type of resident immune cell that play a crucial role in the body’s immune responses, particularly in the context of allergic reactions and other inflammatory conditions. They contain granules rich in histamine and heparin, substances that once released can provoke significant inflammatory responses [28]. Thus, mast cells that are present in the choroid, but not in the retina, may potentially offer a primary cellular target for modulation of angioinflammatory processes affecting choriocapillaris function during AMD pathogenesis. 

Study of choroidal vasculature and cellular components including mast cells, especially in pigmented animals, has been very limited and technically challenging due to the presence of the pigmented layer of RPE cells. We recently reported the development of a method for visualization of choroidal vasculature and innate immune cells in pigmented mouse eyes, thus overcoming the challenge posed by the pigment in the RPE layer. Employing cell-type-specific markers, we have successfully identified choroidal macrophages, mast cells, and the vasculature [29]. We also discovered, for the first time, an arterial circle around the optic nerve resembling the Zinn-Haller arterial circle previously reported in humans and primates. We also noted three distinct choriocapillaris structures, with choroidal mast cells found to be concentrated around the optic nerve. The distribution of mast cells varied among different mouse strains [29]. These methods will enable us to further evaluate the integrity and function of mast cells in various mouse models of AMD. Furthermore, they will help to advance our understanding of the role of mast cells and choriocapillaris dysfunction in pathogenesis of AMD and identification of potential targets for intervention. We will next review what is known about choroidal mast cells and their potential role in maintaining angioinflammatory homeostasis in the choroid.

### 2.1. Choroidal Mast Cells and Angioinflammatory Processes

Mast cells, as crucial effector cells of the innate immune system, have emerged as significant contributors to various inflammatory and autoimmune diseases [30,31]. Choroidal mast cells are strategically located in close proximity to the choroidal vasculature, enabling their active involvement in modulation of angioinflammatory processes. Mast cells are known for their role in allergic reactions, but their function extends beyond this classic role. They are considered as extrahepatic producers of complement proteins and express various complement receptors, including those for C3a and C5a [32]. The complement system, which enhances the ability of antibodies and phagocytic cells to eliminate microbes and damaged cells, engages with mast cells in an intricate manner. The C3a and C5a not only trigger mast cells expressing C3aR and C5aR, but also act as chemoattractants for mast cells originating from various tissue locations [31].

Mast cells also participate in regulating tissue healing and fibrosis, processes that are crucial for repairing injured tissues. However, they can lead to pathological outcomes if not properly controlled [33]. Type 2 immunity, characterized by the production of interleukin (IL)-4, IL-5, IL-9, and IL-13, is frequently observed in tissues during allergic inflammation or infection with helminth parasites. Many pivotal cell types linked with type 2 immune responses, including mast cells, contribute to tissue repair after injury [33]. Mast cells in cancers are found to exhibit a continuous phenotypic expansion specific to the tumor microenvironment [34]. Thus, mast cells can adjust their features in reaction to specific conditions within their microenvironment, potentially influencing disease progression and shaping the response to immunotherapy. In this way, mast cells perform a multifaceted part within the immune system, contributing to inflammatory responses, tissue repair, and the immune response. Understanding the precise mechanisms by which choroidal mast cells contribute to these processes during AMD will have significant clinical importance.

### 2.2. Mast Cells in Retinal Inflammation and Oxidative Stress

Although the retina lacks resident mast cells, systemic activity of mast cells is shown to play a significant role in retinal inflammation and oxidative stress, and pathogenesis of various ocular diseases including retinopathy of prematurity [35]. As indicated, mast cells are a component of the innate immune system and are known for their role in allergic reactions and anaphylaxis. They are also recognized as key players in the regulation of inflammation and immune responses under various pathological conditions. In the context of retinal inflammation, mast cells are involved in the release of pro-inflammatory mediators such as histamine, proteases, cytokines, and chemokines. They produce reactive oxygen species (ROS), which enhance oxidative stress. Mast cells are activated by a multitude of signals, including those from the complement system, cytokines, and toll-like receptors (TLRs). Activated mast cells release a variety of mediators, which contribute to inflammation and oxidative stress. 

Clare et al. [36] reported that mast cell activity contributes to the pathogenesis of retinal diseases by promoting inflammation and oxidative stress. They found that mast cells can be activated by oxidative stress, leading to the release of pro-inflammatory mediators. Wang et al. provided a comprehensive overview of the role of S100A8/A9, a heterodimer expressed in neutrophils and monocytes, in inflammation. S100A8/A9 is released during inflammation and plays a critical role in modulating inflammatory responses by stimulating leukocyte recruitment and inducing cytokine secretion. The authors also suggested that S100A8/A9 could be a potential therapeutic target for inflammation-associated diseases, including retinal inflammation [37]. Rübsam et al. discussed the role of inflammation in diabetic retinopathy, a common complication of diabetes that can lead to blindness. The authors highlighted the involvement of various inflammatory mediators, including those produced by mast cells, in the pathophysiology of diabetic retinopathy [38]. 

One of the key signaling pathways involved in mast cell activation is the high-affinity IgE receptor (FcεRI) signaling pathway. Cross-linking of FcεRI by antigen-bound IgE leads to the activation of Src family kinases, which in turn activate downstream signaling molecules such as Syk, phosphoinositide 3-kinase (PI3K), and mitogen-activated protein kinases (MAPKs). These signaling events lead to the release of preformed inflammatory mediators from mast cell granules, as well as the synthesis of new mediators [39]. In retinopathy of prematurity and diabetic retinopathy, mast cell activation contributes to retinal vascular leakage and neovascularization [35], and systemic inhibition of mast cell activation is protective. 

In AMD, choroidal resident mast cells could promote inflammation and neovascularization through the release of various cytokines and angioinflammatory factors. They can contribute to oxidative stress through the release of ROS and other pro-oxidant mediators [40]. In addition, oxidative stress can further activate mast cells, creating a vicious cycle of inflammation and oxidative damage. Our knowledge regarding the role of resident choroidal mast cells in pathogenesis of AMD is very limited. Delineating the molecular mechanisms and signaling pathways impacted by mast cells is crucial for the development of effective therapeutic strategies for retinal and choroidal diseases. Current treatments for these diseases are limited and mainly involve the management of risk factors and supportive care. However, advances in our understanding of the role of mast cells in inflammation and oxidative stress could pave the way for the development of novel therapeutic approaches. Although current studies highlight the crucial role of mast cells in tissue inflammation and oxidative stress, more research is needed to fully appreciate the true impact of mast cells on these processes, and to develop effective therapeutic strategies targeting mast cells.

### 2.3. Mast Cells in the Choroid

AMD is a progressive retinal disease that leads to the loss of central vision. It is characterized by the accumulation of drusen (extracellular deposits) between RPE and Bruch’s membrane, leading to RPE and photoreceptor cell death [41]. Mast cells are known to be involved in inflammatory responses and are found in various tissues throughout the body, including the choroid. They are well recognized as part of the innate immune system and for their role in allergic reactions. Mast cells contain granules packed with inflammatory mediators like histamine and various proteases, which are released upon activation [42]. This release, known as degranulation, rapidly provides a variety of inflammatory mediators into the surrounding tissue. These mediators can cause vasodilation, recruit other immune cells to the site of inflammation, and stimulate the production of cytokines and chemokines, further enhancing inflammation.

In AMD, chronic inflammation and oxidative stress contribute to the degeneration of the RPE and the underlying choroidal vasculature. The resident choroidal mast cells, through their release of inflammatory mediators, could contribute to this chronic inflammation and oxidative stress associated with pathogenesis of AMD [27]. Furthermore, mast cells are involved in the activation of the complement system. Although dysregulation of the complement system is implicated in the pathogenesis of AMD [43], the underlying mechanisms remain unknown. Choroidal mast cells can release proteases that activate the complement system, potentially contributing to complement dysregulation in AMD [31]. 

Ogura et al. reported that mast cells are abundantly present in the choroid and their continuous stimulation and degranulation, particularly through mast-cell-derived tryptase, could be central to the progression of GA in a preclinical model. These studies demonstrated that the degeneration of RPE cells and retinal and choroidal thinning, which are characteristics of GA, were driven by chronic stimulation and activation of choroidal mast cells. They showed interventions targeting mast cell degranulation or inhibiting tryptase activity could prevent the development and progression of the disease phenotypes [26]. Thus, it is reasonable to propose that choroidal mast cells, through their role in inflammation, oxidative stress, and complement activation, are key contributors to the pathogenesis of AMD. 

### 2.4. Choroidal Mast Cell Activation in AMD

The implications of choroidal mast cell activation in AMD are profound and multifaceted, with recent research beginning to elucidate the complex interplay between choroidal mast cells, RPE cells, and pathogenesis of dry AMD [25]. Activated choroidal mast cells release various mediators such as histamine, cytokines, and proteases [28,44]. In the context of dry AMD, choroidal mast cell activation is linked to the degeneration of the RPE and photoreceptors in preclinical models. The RPE cells are essential for visual function, and their degeneration is a hallmark of dry AMD. Recent in vitro studies show that mast cells induce RPE cell death by release of granules containing tryptase, a potent protease. The signaling pathways involved in this process are complex and involve multiple steps. Upon activation, mast cells release tryptase, which cleaves and activates protease-activated receptor-2 (PAR-2) on the surface of RPE cells. PAR-2 activation triggers a signaling cascade involving the phosphorylation of the MAPKs including ERKs1/2 and JNK, leading to the activation of the transcription factor AP-1. AP-1 induces the expression of the death receptor FasL, which binds to its receptor Fas on the same cell, triggering apoptosis [25,45]. Importantly, this process seems to be exacerbated by oxidative stress, a condition known to contribute to the pathogenesis of dry AMD. Oxidative stress can enhance the release of tryptase from mast cells, further promoting RPE cell death [46].

In addition to their direct effects on RPE cells, choroidal mast cells could also contribute to dry AMD through their interactions with other immune cells. For instance, mast cells could recruit and activate macrophages, which release additional pro-inflammatory and cytotoxic factors, potentially exacerbating RPE damage [44]. Thus, choroidal mast cell activation could play a significant role in the pathogenesis of dry AMD, primarily through the induction of RPE cell death. This process involves complex signaling pathways and is influenced by factors such as oxidative stress. However, whether this occurs in vivo and the role changes in TSP1 levels play remain unknown. We propose the mouse sodium iodate (NaIO_3_) acute model of outer retinal degeneration is a suitable model to explore these possibilities. 

## 3. The Sodium Iodate (NaIO_3_) Acute Model of AMD

Sodium iodate (NaIO_3_)-mediated outer retinal degeneration is a well-established, reproducible acute model for studying many aspects of dry AMD pathogenesis. NaIO_3_ treatment induces RPE degeneration and subsequent photoreceptor cell death, mimicking the pathophysiological changes observed in dry AMD [47]. The mechanism of NaIO_3_-induced outer retinal degeneration is primarily attributed to the generation of reactive oxygen species (ROS) and oxidative stress, which lead to cell death [47,48]. However, the details of these mechanisms and their impact on TSP1 expression, mast cell activation, and choriocapillaris dysfunction remain largely obscure.

Intracellular oxidative stress, including the production of cytosolic H_2_O_2_ and mitochondrial superoxide, is promoted by NaIO_3_, which is accompanied by the activation of mitochondrial biogenesis and autophagy, which is specifically responsible for the removal of damaged mitochondrial components [49]. The NaIO_3_ treatment of RPE cells results in upregulation of phosphorylated (p)-Nrf2, PGC1α, and sirtuin 1 (Sirt1). In addition, treatment with quercetin downregulated the levels of p-Nrf2, heme oxygenase 1 (HO1), Sirt1, and PGC1α, influencing the acetylation of superoxide dismutase 2 (SOD2) [49]. These results suggest that quercetin could have potential therapeutic efficacy for the treatment of AMD through regulation of mitochondrial ROS homeostasis by deacetylation of SOD2. Thus, the NaIO_3_ model of outer retinal degeneration provides a valuable tool for studying the pathophysiological changes associated with dry AMD, including oxidative stress, mitochondrial dysfunction, and autophagy. The use of antioxidants such as quercetin may offer a potential therapeutic approach for mitigating these changes, and likely prevention of dry AMD pathogenesis. However, the contribution of changes in TSP1 levels and choroidal mast cell activation by NaIO_3_ remains unexplored. 

### 3.1. NaIO_3_-Induced Oxidative Stress and Inflammation

As indicated, the NaIO_3_ acute model of outer retinal degeneration mimics some of the pathological features observed in dry AMD, including RPE cell death, photoreceptor degeneration, and inflammation [31]. In the context of NaIO_3_-induced oxidative stress and inflammation, recent studies have shown that NaIO_3_ triggers considerable RPE cell death and decreases cellular metabolic activity. This oxidative stress also leads to the activation of the alternative complement pathway. Specifically, oxidative stress downregulates the expression of CFH and upregulates the expression of complement components C3a and C5a [50]. CFH is a critical regulator of the alternative complement pathway, acting as a negative regulator protecting against complement activation. Under oxidative stress conditions, the expression of CFH mRNA and protein is reduced, leading to an increased activity of the alternative complement pathway [51]. In addition, C3a and C5a, fragments of C3 and C5, respectively, act as anaphylatoxins and play a distinct role in pathogenesis of AMD. These proteins promote the recruitment and activation of phagocytic immune cells, and likely choroidal mast cells, to sites of tissue damage and the production of proinflammatory cytokines, leading to local chronic inflammation [52].

Recent studies have shown that certain treatments can modulate the expression of these complement components under oxidative stress conditions. For instance, Qihuang Granule (QHG), a traditional Chinese medicine, significantly improves the expression of CFH mRNA and upregulates CFH protein expression in RPE cells, mouse sera, and mouse retinas. Moreover, QHG significantly decreased the H_2_O_2_- and NaIO_3_-induced increases in the mRNA and protein expression of C3a and C5a in RPE cells and mouse serum and retinas, respectively [53]. These findings suggest that QHG has antioxidative and anti-inflammatory effects, and it protects RPE cells from oxidative stress via regulation of the alternative complement pathway, which could halt or delay disease progression in AMD. However, the contribution of changes in TSP1 levels and mast cell activity to these events remains unknown.

Further studies have elucidated the roles of the P2X7 receptor (P2X7R), a receptor for extracellular ATP, in NaIO_3_-induced oxidative stress and inflammation and outer retinal degeneration. P2X7R is a ligand-gated ion channel that is expressed in various retinal cell types, including microglia, photoreceptors, RPE, and Müller glial cells. Its expression can be upregulated during vitreoretinopathy. P2X7R activation induces cell death in these retinal cells, contributing to retinal degeneration [54]. In addition, the P2X7R/NLRP3 pathway plays a crucial role in promoting inflammation, pyroptosis, and apoptosis of retinal endothelial cells in the early stages of diabetic retinopathy. The nucleoside reverse transcriptase inhibitor, 3TC, effectively mitigates these effects by targeting P2X7R large pore formation, reducing inflammation, apoptosis, and pyroptosis in both in vitro and in vivo models, providing new insights into the mechanism of 3TC in diabetic-environment-induced retinal injury [54].

In a study, P2X7R activation by a selective agonist, 2′(3′)-O-(4-Benzoylbenzoyl)ATP (BzATP), was found to lead to cytotoxicity in a cell-type-specific manner, with a preference for microglia [55]. This effect is correlated to their relative P2X7R expression level. Among the four cell types, BV-2 (microglial cell line) expressed a much higher level of P2X7R than 661 W (photoreceptor cell line), rMC1 (muller cell line), and RPE cells [55]. However, NaIO_3_ can cause a similar degree of cell death in all these cell types. BzATP can also increase the cytotoxicity of NaIO_3_ only in ARPE-19 cells. This effect might be due to the upregulation of P2X7R expression in NaIO_3_-treated RPE cells, as well as other unclarified integration of intracellular death events in RPE cells [54]. We have found that mast cells also express a P2X7R receptor. However, the impact of its activation in choroidal mast cell activity and the TSP1 level remains unknown. Thus, P2X7R plays a significant role in NaIO_3_-induced oxidative stress and inflammation in outer retinal degeneration, and its modulation could be a potential therapeutic strategy for dry AMD.

### 3.2. NaIO_3_, RPE Cell Damage, and Its Implications in Dry AMD

NaIO_3_ is an oxidative toxic agent that selectively damages RPE cells, making it a useful tool for creating reproducible in vitro and in vivo models to study some aspects of AMD-related pathogenesis [56,57,58]. Although NaIO_3_ is not directly involved in AMD pathology, it provides a means to understand the mechanisms of RPE cell death and likely choriocapillaris degeneration as key pathogenic markers of AMD [48]. Recent studies have shown that NaIO_3_ induces oxidative damage and promotes apoptosis in RPE cells [59,60], characterized by decreased cell viability, increased ROS levels, increased expression of pro-apoptotic proteins such as Bax and cleaved caspase-3 [61,62,63,64], and decreased Bcl-2 expression [62]. Interestingly, a peptide derived from a highly conserved region of the human lens protein αA-crystallin, known as mini-αA, has a protective effect against NaIO_3_-induced outer retinal degeneration [65,66]. Treatment with mini-αA increases cell viability, reduces ROS levels, and modulates the expression of Bax, cleaved caspase-3, and Bcl-2, effectively reversing the oxidative damage and apoptosis induced by NaIO_3_ [67]. Our preliminary studies indicate that NaIO_3_ can suppress TSP1 expression in the mouse choroid and induce mast cell activation. However, the direct role of these changes in outer retinal degeneration in a NaIO_3_ model needs investigation.

Additional studies have revealed a potential role for microRNA-155-5p (miR-155-5p) in NaIO_3_-mediated RPE cell loss. The miR-155-5p was found to be significantly upregulated upon NaIO_3_ treatment and downregulated after mini-αA treatment, suggesting that it plays an inhibitory role in RPE cell apoptosis [68]. Bioinformatics prediction and functional validations have identified cyclin-dependent kinase 2 (CDK2) as a target gene for miR-155-5p. CDK2 is an important regulator of the cell cycle, and its expression is modulated by miR-155-5p in the context of NaIO_3_-induced oxidative damage [69]. Thus, NaIO_3_-induced oxidative stress and apoptosis in RPE cells is a useful model for studying AMD pathogenesis. The protective effects of mini-αA and the regulatory role of miR-155-5p provide valuable insights into the molecular mechanisms underlying RPE cell death.

Zhang et al. investigated the effect of NaIO_3_ on RPE cell death and found that NaIO_3_ induced calpain-2 activation, autophagy, lysosomal dysfunction, and cell apoptosis [57]. Calpain-2 is a calcium-dependent, non-lysosomal cysteine protease that plays a role in various cellular processes, including cell motility, cell cycle progression, apoptosis, and synaptic plasticity. They found that NaIO_3_-exo, exosomes derived from RPE cells under NaIO_3_ stimulation, induced calpain-2 activation, which in turn led to autophagy and cell apoptosis. The increase in autophagic substrates was related to lysosomal dysfunction. Furthermore, NaIO_3_-exo-activated calpain-2 participated in the autophagy–lysosomal pathway (ALP) function and apoptosis in RPE cells [57]. This study also found that NaIO_3_-exo enhanced calpain-2 expression, ALP dysfunction, apoptosis, and retinal damage in rats. These results are in accordance with the in vitro data. Collectively, these studies provide a mechanistic link between calpain-2 and impaired ALP function and identify calpain-2 as a promising molecular target for AMD therapy [57].

Necroptosis is a form of programmed necrosis or inflammatory cell death, distinct from apoptosis. It is characterized by the swelling of organelles, plasma membrane rupture, and the release of intracellular components, leading to inflammation. Recent studies suggest that necroptosis may play a role in the pathogenesis of AMD [70], specifically in the advanced forms of the disease characterized by geographic atrophy or disciform scarring. Exposure to NaIO₃ can activate pathways leading to RPE cell death, including necroptosis. When RPE cells undergo necroptosis, they release damage-associated molecular patterns (DAMPs) that can activate immune responses and contribute to further tissue damage, creating a vicious cycle of inflammation and degeneration [71]. The receptor-interacting protein kinase (RIPK)-dependent necrotic pathway is pivotal for RPE and photoreceptor cell death in AMD. RIPK-dependent necroptosis, especially under oxidative stress, intensifies neuroinflammation. Ripk3 deficiency in the dry AMD model reduced the release of DAMPs and suppressed the inflammatory response in the retina [72]. Understanding the relationship between NaIO₃-induced necroptosis and AMD provides insights into the molecular mechanisms driving RPE degeneration. Hanus et al. confirmed that RPE necroptosis is the predominant mechanism of NaIO_3_-induced RPE cell death [70]. By targeting specific molecules, or pathways in the necroptotic processes, there might be potential for developing novel therapeutic strategies for AMD [73]. For instance, necrostatin-1, an inhibitor of necroptosis, might offer protective effects against NaIO₃-induced RPE cell death and, by extension, in AMD [74].

The role of ferroptosis, a regulated cell death process driven by iron-dependent lipid peroxidation, in RPE cell death has been recently explored [75,76]. Ferroptosis, an iron-dependent form of regulated cell death, has been proposed as a potential mechanism behind RPE cell death in AMD, especially given the prominence of oxidative stress in the disease’s pathology. NaIO₃-induced ferroptosis in RPE cells serves as a tangible model to study this connection. The iron-dependent lipid peroxidation, characteristic of ferroptosis, offers insights into how oxidative damage could lead to cell death in the RPE, exacerbating AMD’s progression [77]. NaIO_3_ triggers ferroptosis in RPE cells by oxidizing cysteine and GSH, leading to increased iron and ROS levels; however, these in vitro findings might not directly translate in vivo. A study by Gupta et al. found that lipocalin 2 (LCN2), a protein that is upregulated in the RPE of a mouse model of dry AMD, decreases autophagy and activates the inflammasome–ferroptosis processes [78]. This suggests a potential link between NaIO_3_-induced RPE cell damage and ferroptosis, although further studies are needed to fully elucidate this relationship. Further evaluations are needed in rodent models to understand NaIO_3_’s effects on retinal degeneration and the potential benefits of ferroptosis inhibitors [48,79].

### 3.3. Therapeutic Targets in NaIO_3_-Induced Outer Retinal Damage

We already discussed the significance of oxidative stress with NaIO_3_ treatment and the use of antioxidants as potential protection. One of the key signaling pathways implicated in NaIO_3_-induced RPE cell damage is the MAPK pathways [48]. These pathways are involved in a variety of cellular processes, including cell proliferation, differentiation, and apoptosis. In the context of dry AMD, activation of the MAPK pathways has been linked to increased oxidative stress and inflammation, leading to RPE cell death [48,80]. Another important target is the PI3K/AKT pathway, which is involved in cell survival and proliferation. Dysregulation of this pathway has been associated with increased oxidative stress and apoptosis in RPE cells following NaIO_3_ exposure. In addition to these signaling pathways, several key molecules have been identified as potential therapeutic targets. These include TNFα, a pro-inflammatory cytokine that is upregulated in response to NaIO_3_-induced oxidative stress, and VEGFA, a growth factor involved in angiogenesis that is also upregulated following NaIO_3_ exposure. Furthermore, recent studies have highlighted the role of autophagy in NaIO_3_-induced RPE cell damage [81]. Thus, the damage caused by NaIO_3_ involves multiple signaling pathways and molecular targets, including the MAPK and PI3K/AKT pathways, TNFα, VEGFA, and autophagy-related molecules. The role of these pathways and changes in TSP1 levels, and likely choroidal mast cell activation, in AMD deserves future consideration.

## 4. Thrombospondin-1 (TSP1) and Pathogenesis of AMD

Thrombospondin-1 (TSP1) is a multifunctional protein that plays a significant role in tissue remodeling and inflammation. Recent studies have shed light on the complex mechanisms through which TSP1 influences these processes. TSP1 is a matricellular protein that interacts with various cell surface receptors, cytokines, and growth factors, thereby modulating cell behavior and tissue physiology. TSP1 has been implicated in the regulation of cell proliferation, apoptosis, and migration, all of which are critical processes in tissue remodeling [82].

In the context of inflammation, TSP1 modulates the immune response by influencing the activity of various immune cells. For instance, TSP1 can suppress the activation of T cells, thereby modulating the immune response. It also binds to and activates latent transforming growth factor-beta (TGF-β), a cytokine that plays a crucial role in inflammation and tissue remodeling [83,84]. Recent studies in the kidney have revealed the role of TSP1 in tissue remodeling and inflammation. TSP1 expression in the kidney can promote fibrosis, a form of tissue remodeling characterized by excessive deposition of extracellular matrix proteins. TSP1 could induce fibrosis by activating TGF-β and promoting the differentiation of fibroblasts into myofibroblasts, cells that play a key role in fibrosis [85,86,87].

Another study showed that TSP1 can modulate inflammation by influencing the behavior of macrophages, immune cells that play a crucial role in inflammation. The study found that TSP1 can promote the polarization of macrophages toward a pro-inflammatory phenotype, thereby enhancing the inflammatory responses [88]. Furthermore, TSP1 has been implicated in the regulation of angiogenesis, the formation of new blood vessels from pre-existing capillaries. This process is critical in tissue remodeling and is often dysregulated under pathological conditions characterized by chronic inflammation. TSP1 can inhibit angiogenesis by binding to and activating CD36, a receptor expressed on the surface of endothelial cells [89]. Activation of CD36 by TSP1 triggers a signaling pathway that leads to the apoptosis of endothelial cells, thereby inhibiting angiogenesis [90]. Understanding the complex roles of TSP1 in tissue remodeling and inflammation could provide valuable insights into the pathogenesis of AMD and the development of novel therapeutic strategies.

### 4.1. Dysregulation of TSP1 in Pathogenesis of AMD

Dysregulation of TSP1 has been previously implicated in the pathogenesis of neovascular or wet AMD [91,92,93,94]. The development and progression of wet-AMD are mainly attributed to aberrant neovascularization. A pivotal component of this pathological process is choroidal neovascularization (CNV), where new blood vessels sprout from the choroid layer beneath the RPE into the subretinal space [95]. One of the molecular actors implicated in modulating this neovascularization process is TSP1. TSP1 can inhibit angiogenesis by directly influencing endothelial cell migration and proliferation, and by modulating the activity of matrix metalloproteinases (MMPs), enzymes that degrade the extracellular matrix and facilitate angiogenesis. Dysregulation of TSP1, therefore, can lead to aberrant angiogenesis, contributing to the progression of AMD [82]. TSP1 interacts with a variety of cell surface receptors, including CD36 and CD47, to modulate endothelial cell migration, proliferation, survival, and capillary morphogenesis.

In addition to its role in angiogenesis, TSP1 is also involved in the regulation of inflammation, a key factor in the pathogenesis of both the dry and wet forms of AMD [83,93,94]. TSP1 can modulate the activity of various immune cells, including macrophages and T cells, and influence the production of inflammatory cytokines [96,97]. TSP1 expression is recently reported to suppress mast cell activation [98]. Thus, dysregulation of TSP1 can lead to chronic inflammation, a condition that is thought to contribute to the degeneration of retinal cells in AMD [82,99]. Furthermore, TSP1 has been shown to interact with various signaling pathways that are implicated in AMD. For instance, TSP1 can activate transforming growth factor-beta (TGF-β), a signaling molecule associated with fibrosis and scarring in the late stages of AMD [86,100,101,102,103]. TSP1 can also interact with the CD36 receptor, which is involved in the clearance of drusen, deposits that accumulate in the retina of patients with AMD [104,105]. However, the molecular mechanisms underlying TSP1 regulation in CNV remain incompletely understood. Emerging evidence points to a complex interplay of genetic, epigenetic, and environmental factors. For instance, hypoxia, oxidative stress, and inflammation—conditions associated with wet AMD—have been shown to regulate TSP1 expression [106].

TSP1 plays a multifaceted role in the pathogenesis of AMD, influencing processes such as angiogenesis, inflammation, and various signaling pathways. Dysregulation of TSP1, therefore, can contribute to the development and progression of AMD. We previously showed mice deficient in TSP1 exhibit increased inflammation and CNV in the mouse laser model [93]. In addition, a mimetic peptide from TSP1 mitigates CNV in this model. Further research is needed to fully elucidate the role of TSP1 in AMD and to explore the potential role of TSP1 as a modulator of mast cell homeostasis and regulation of angioinflammatory processes in the choroid (Figure 1).

### 4.2. Therapeutic Potential of TSP1 in AMD

In the expanding realm of research focused on AMD, significant effort has been dedicated to comprehending the biological elements that affect the emergence and advancement of this prevalent cause of elderly vision impairment. Among these elements, TSP1 has been gaining growing acknowledgment due to its crucial involvement in regulation of neovascularization and inflammation in the choroid and a target for treating AMD. As an essential anti-angiogenic glycoprotein, TSP1 interacts with several cell surface receptors, modulating vital processes such as cell proliferation, migration, and survival, and can inhibit abnormal blood vessel formation [82]. This role implies that TSP1 could be harnessed to manage the deviant neovascularization characteristic of wet AMD, an advanced manifestation of the condition.

Evidence supporting the therapeutic potential of TSP1 in AMD has been demonstrated through both observational and interventional studies. Notably, studies have shown that decreased expression of TSP1 in the RPE and choroid correlates with wet AMD progression, suggesting that TSP1 deficiency may contribute to disease pathogenesis [92,93]. Despite the potential highlighted by these initial inquiries, several hurdles exist in the therapeutic utilization of TSP1 in AMD. While excessive TSP1 appears to mitigate CNV, its pleiotropic actions might lead to undesired effects. Thus, pinpointing the ideal dosage and delivery method to achieve desired outcomes while limiting side effects is imperative [107]. Thus, we propose TSP1 holds significant promise as a therapeutic target in AMD. However, more comprehensive preclinical and clinical investigations are essential to discern the safety, efficacy, and optimal application of TSP1-focused interventions. As our comprehension of AMD’s intricate biology deepens, the evolution of targeted therapies provides optimism for improved management of this debilitating ailment [108].

## 5. The Interplay between Oxidative Stress, Mast Cells, and TSP1

The potential interplay between these entities paints a complex picture of dry AMD’s etiology. Preliminary research suggests that the RPE damage induced by oxidative stress could be a trigger for mast cell activation and infiltration [109]. The activated mast cells might then release mediators that further deteriorate RPE cell function [55,109,110] and modulate the expression or activity of TSP1, thus exacerbating the disease’s progression [83]. In addition, TSP1 might modulate mast cell functions, representing a potential therapeutic angle [98]. Furthermore, the altered TSP1 milieu, in turn, might amplify the pro-inflammatory actions of mast cells, creating a feedback loop driving retinal damage. The intricate interplay among oxidative stress, mast cells, and TSP1 holds significant importance in our understanding of dry AMD’s pathophysiology. Unraveling these connections could lead to innovative therapeutic strategies targeting specific components of this triad.

### Potential Therapeutic Avenues Targeting This Interplay

The intricate dynamics between mast cells and TSP1 within the choroidal environment and their potential implications for AMD present an intriguing platform for therapeutic development. Here, we delineate the therapeutic avenues that could capitalize on this molecular interplay. Given the role of mast cells in inducing oxidative stress and inflammation, agents that stabilize mast cells can be repurposed for dry AMD treatment. Mast cell stabilizers, such as ketotifen fumarate, which are primarily used in allergic conditions, may prevent the release of mast cell mediators and thus curb inflammation and oxidative stress [26,111].

Developing molecules that can mimic the actions of TSP1, especially its anti-inflammatory and anti-angiogenic properties, can also be a potential therapeutic strategy. These mimetics could either enhance the endogenous properties of TSP1 or prevent its degradation, thereby ensuring its protective roles in choroidal environments [93]. We have shown a TSP1 mimetic peptide (ABT898) inhibits inflammation and CNV in a preclinical model of wet AMD [93]. However, whether this peptide has any impact on choroidal mast cell activation and outer retinal degeneration in dry AMD awaits future investigation. 

In addition, considering that both mast cells and TSP1 modulate inflammatory pathways, drugs like corticosteroids or newer-generation non-steroidal anti-inflammatory agents could be explored for their efficacy in reducing inflammation, a key driver of AMD [112]. Given that TSP1 has an integral role in cell–matrix interactions, and that mast cells can influence the integrity of structures like Bruch’s membrane, agents that modulate cell–matrix interactions or prevent matrix degradation might also prove beneficial. These include matrix metalloproteinases and mast cell protease inhibitors, which might prevent degradation and maintain the integrity of the choroidal environment [113,114]. 

A combined approach targeting both mast cell mediators and modulating TSP1 functions might offer a comprehensive therapeutic strategy. Such regimens could be tailored based on the stage of AMD or individual patient profiles, allowing for personalized treatments [115,116]. Emerging technologies in the realm of gene therapy could also be exploited to modulate the expression of TSP1 or genes associated with mast cell activation. By targeting the genetic basis of this interplay, long-term modulation and potentially even reversal of AMD progression might be feasible [117]. Utilizing nanotechnology for targeted drug delivery can enhance the specificity and reduce systemic side effects. Nano-carriers can be designed to deliver drugs specifically to target cells, modulating the interplay between mast cells and TSP1 at the cellular level [118].

## 6. Conclusions

Mast cells, predominantly recognized for their roles in immune responses and allergies, have recently been identified as pivotal players in outer retinal degeneration, especially AMD. Choroidal mast cells, with their unique position in the choroid and their dynamic release of bioactive mediators, are likely active participants in the progression of inflammatory and degenerative conditions [12,26]. Other studies further solidify this viewpoint by elucidating the mast cell’s potential in exacerbating oxidative stress, inflammation, and cellular damage in the ocular environment [27,36,119].

Ma et al. argued for the urgency of more nuanced studies centered on the specific dynamics and mechanistic roles of mast cells in the outer retina, emphasizing the necessity to understand their intricate interactions with other ocular cells and molecular entities [45]. A deep dive into their genetic, molecular, and cellular pathways, along with their interactions with other cellular components, can shed light on the comprehensive etiology of retinal degenerative processes.

Given the potential integral role of choroidal mast cells in AMD progression, they may emerge as promising targets for therapeutic interventions. Recent studies outline potential strategies that pivot around mast cell modulation, which could pave the way for more effective treatments for AMD [120]. These insights suggest that therapies aiming at mast cell stabilization or modulation of their mediator release could potentially curb AMD’s progression [12]. Furthermore, as our understanding deepens, there exists an exciting avenue for the development of personalized treatments targeting specific mast cell pathways, thereby allowing for more precise and patient-centric therapeutic strategies.

In summary, understanding the role of choroidal mast cells in outer retinal degenerations, and more specifically in AMD, is reshaping our perception of AMD pathogenesis. Their significance not only offers insights into the disease’s mechanistic pathways but also will open up innovative therapeutic avenues. As research progresses, the therapeutic potential stemming from choroidal mast cell modulation in AMD promises to be a critical area of focus.

## Figures and Tables

**Figure 1 cells-13-00050-f001:**
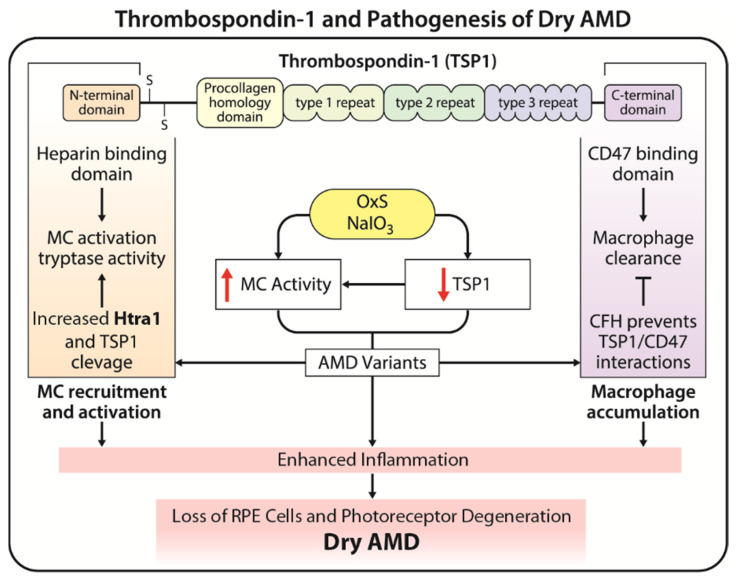
Thrombospondom-1 (TSP1) is a key regulator of ocular angioinflammatory processes. We propose modulation of TSP1 levels by oxidative stress and/or AMD gene variants leads to enhanced mast cell recruitment and activation, inflammation, loss of RPE and photoreceptor cells, degeneration of choriocapillaris, and resulting dry AMD [12].

## Data Availability

All the data presented here are included in the manuscript. Further inquiries should be directed to the corresponding author.

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
