# Peer review of "Choroidal Mast Cells and Pathophysiology of Age-Related Macular Degeneration"

_cells, 2023, doi:10.3390/cells13010050_

Round 1
Reviewer 1 Report
Comments and Suggestions for Authors
acceptable after minor editing
Reviewer 2 Report
Comments and Suggestions for Authors
· The manuscript “Choroidal Mast Cells and Pathophysiology of Age-Related Macular Degeneration” by Sara Malih et al. seeks to highlight the involvement of choroidal mast cells in the pathogenesis of Age-related Macular Degeneration (AMD). We agree with the authors that the contribution of mast cells to AMD biology is under reported. However, the authors miss several key points that, if incorporated, would strengthen the discussion, but without, greatly reduces enthusiasm for and significance of the manuscript.
· Primary among these is the racial bias of AMD incidence and progression, which could be coupled to the racial variation of mast cell activity. Based on the title, I expected racial variation in mast cell activity to be linked to AMD incidence and progression. Probably interesting. Unfortunately, the manuscript is repetitive, lacks focus, lacks the depth in the discussion of AMD risks, and fails to develop new concepts for previous observations. Significant weaknesses.
· For example, discussion of association of compliment factor H (CFH) as a risk factor for AMD is lacking important detail. This is one of the most studied genes linked to AMD. The authors note “A common polymorphism in the CFH gene (Y402H) has been strongly associated with an increased risk of developing AMD”. This only partially true, and directly reflects a central problem in the manuscript. The authors completely miss the fact that CFH allele Y402H is only a risk factor in certain ethnicities and populations. In fact, study of segregated Asian and Chinese populations suggests the opposite for the allele, a lack risk from the CFH Y402H allele, and in fact, in one study, the risk varied significantly by race and location or ethnicity (see PMC3359047). This is a review article, and as such, needs to be a complete, unbiased, and deliver an accurate view of today’s data and interpretations. Depending which study I look at, I could make the case that CFH Y402H is protective from AMD incidence in some Asian groups. This manuscript simply states it’s a risk factor, which is not universally true, and the authors should present the complete and accurate picture. They did not. Think of this, Y402H is only sometime a risk factor? If it has a a functional consequence and drives AMD incidence, then why wouldn't that be universal?
· Sodium Iodate toxicity is not a model of AMD. It is a model of RPE toxicity, but it does not mimic the AMD disease process. AMD is not simply RPE cell death, so any potential relationship to RPE iodate immediate toxicity is difficult to postulate. RPE don’t just die in AMD. Why is this here, and how does it relate to choroidal mast cells? There is nothing to really tie mast cells, iodate toxicity, and AMD together.
· The use of data from ARPE-19 cells needs to consider that the cells are transformed, proliferative, and unpigmented. So, they lack several primary properties human RPE likely important in AMD. Simply put, ARPE-19 are not a good model for RPE activity in AMD pathology. Further, can you relate ARPE-19 and sodium iodate to the title of the manuscript? Mast cells? I don’t see it.
· Summary. The absolute lack of any incorporation of race, pigmentation, and ethnic diversity - each which likely impacts mast cell function and AMD biology, greatly reduces any enthusiasm for this submission. The inclusion of iodate and ARPE-19 detracts from the proposed focus of the manuscript. They are not mast cells.
Specific comments:
1. The discussion from lines 72-188 do not include mast cells, which is the subject of the manuscript. Then, the following paragraphs finally describe mast cell function, but these ideas need strong integration, and probably start with mast cells, the primary topic of the manuscript.
2. This idea is repeated “In AMD, choroidal resident mast cells could promote inflammation and neovascularization through the release of various cytokines and angioinflammatory factors.”, but is not well developed. Any cell in the eye 'could' do this. Anything specific for choroidal mast cells? Any data directly implicating mast cells and AMD pathology?
3. Similar to pigmentation, the authors should be clear when discussing pigment, animals, or pigmented RPE cultures, and amelanotic or albino RPE and the animals need to be explicitly identified and clearly developed.
4. AMD is strongly race related and this need to be central. Why, and are mast cells involved?
5. Justify and explain any use of sodium iodate models and results. This is a major weakness.
